# OpenReview forum: "First-Person Fairness in Chatbots"
_ICLR.cc/2025/Conference — ICLR 2025 Spotlight_

### Official Review · Reviewer_rDcM · 2024-11-01

**Soundness:** 3
**Presentation:** 2
**Contribution:** 2
**Rating:** 5
**Confidence:** 4

**Summary:**

This paper introduces a method for evaluating one form of first-person fairness, particularly analysing name-based biases. They apply this approach to chats from public data as well as a commercial chatbot across various small and large models. Gender and racial bias is explored in this study.

The method involves generating a pair of responses from the target model corresponding to each group (for e.g. male and female for gender), and prompting a judge language model (referred to as AI RA) to identify harmful stereotypes for the pair. A harmful-stereotype score is calculated using the output from the AI RA. Additionally, they utilise an existing approach to identify qualitative differences in the response pairs. Their results indicate largest biases in older generations of models and in open-ended tasks.

**Strengths:**

1. This paper quantifies name-based fairness towards the user (of a chatbot) and also introduces harmful-stereotype scores to measure it, both novel contributions.
2. The experiments are quite robust. Evaluation is done using both public and private data and the authors also perform correlation analysis between AI RA and human evaluation.

**Weaknesses:**

1. The paper lacks enough significance since it addresses a very specific issue, name-based fairness for chatbot users, and it only covers binary gender and racial biases. Based on the results, the presence of such stereotypes is also rare, < 0.3 % in Fig. 5, which indicates that this problem may not be as important as general fairness of LLMs. Expanding this work to address multiple aspects of first person fairness is necessary to create a significant impact.
2. The first half of the paper, especially the introduction section, requires additional effort to understand properly. The content is not structured clearly, consists of a few complex sentences and lacks technical depth.
3. The paper also has some formatting issues. There are multiple instances of words being fused together and missing a space for e.g. 'Weextract' on page 4, 'LMSYSand; on page 7, etc. The abstract mentions authors also discuss methods for monitoring and reducing biases, but there is no mention of these points in the main body of the paper. This indicated that the authors didn't review the paper in sufficient detail.

**Questions:**

None

---

### Official Review · Reviewer_uH1B · 2024-11-02

**Soundness:** 3
**Presentation:** 2
**Contribution:** 3
**Rating:** 8
**Confidence:** 4

**Summary:**

The paper studies first person fairness that is fairness towards direct users based on their demographic information. More specifically they study fairness in the scope of LLM users with regards to their names. They consider in person fairness considering race and gender. In addition, they propose a privacy preserving method based on using an LLM as a research assistant (AI RA) to detect existing stereotypical behavior in LLMs against its users.

**Strengths:**

1. The motivation of the paper is clear and the paper tackles an important question.
2. The idea of using an AI research assistant (AI RA) is interesting specially for preserving privacy.

**Weaknesses:**

While the motivation behind the paper is clear and the paper tackles an important problem there are some weaknesses as listed below:

1. The domain is so specific and narrow to biases in names. The paper could have been written considering more general cases on how AI RA can be used in general to study and tackle biases in a privacy preserving manner.


2. The approach is limited to pair-wise comparison which can make the approach limited.


3. For identifying domains and tasks, authors considered various conditions. These conditions overly simplify the problem which might artificially make the AI RA look good. Please address this shortcomings in the ethical statement of the paper. To not mislead the reader on naive over-reliance on the AI RA.


4. It is good that authors have acknowledge the inherent randomness problems and how they might effect the outcome (lines 276-280), however, it would be better if some suggestions are provided on how to improve against these randomness.


5. Scale of the names to study gender bias is small and only limited to US names.


6. Experiments and Results section could improve both by conducting more comprehensive experiments and more detailed discussions.
For instance, one interesting thing that could have been studied in more depth was the persona that AI RA could take and its effects on the results. Authors could have also considered non-binary gender names. Expanding to different types of biases and some techniques to enhance the AI RA and some results and experiments that would showcase that the enhancement was effective.


7. For Table 1, I would suggest to explain the variance in the correlation in detail e.g. why correlation for hispanic names are low and asian high.


8. While the paper strongly claims that the approach can be more privacy preserving, there are no studies on actually verifying it other than the claim that since now AI RA is conducting the study it might be more privacy preserving. It would be good to do some small analysis on this aspect as well.


9. In the ethics statement provide a word or two about AI RA approach and its limitations and perhaps if we over-rely on it with no human intervention.

_________________________
Minor comment: There are many typos and writing issues. Please fix and do a careful proofreading. Examples are as follows:

Line 78: Weuse

Line 102-103: ?/missing citation

Line 193: Weextract

**Questions:**

1. The paper claims that "We find that AI RA ratings closely match human ratings for gender bias, but less so for other biases, such as racial bias." are there any insights on why this is the case? Please address this in the paper as well.

---

### Official Review · Reviewer_Ya41 · 2024-11-03

**Soundness:** 3
**Presentation:** 4
**Contribution:** 2
**Rating:** 8
**Confidence:** 3

**Summary:**

The paper studies the problem of bias in the chat bots with generative AI. The bias is normally introduced when the bot has access to additional information about the users in the chat, when the gender and race biases may come into play. The work presents the method to evaluate the fairness / detect the unfairness of the chatbot towards the user. The authors use the LLMs to assess the fairness on a large scale but there is a human in the loop to confirm that the assessment is correct. The authors claim that gender bias is higher in older generations of models and open ended tasks after their analysis with AIRA and confirmation with the use of human testers.

**Strengths:**

1. The paper communicates well the existing bias in the responses of LLMs to people of different genders.
2. The importance of the first person fairness is explained well in the section 1.2
3. Detailed analysis of performance of AIRA compared to human testers is provided.

**Weaknesses:**

A few comments for authors to consider:
- Possibility to introduce bias and/or unfairness through AIRA itself. The versions that have been tested through cross-validation with human raters might be good to use with the same settings, but the method will not scale to the versions that have not been tested. Thus, with each update and step forward in the LLMs there is a need to run a manual evaluation of the model to see if it is feasible to be used as AIRA. In case the authors believe that future models will not be worse in their abilities to serve as AIRA, this needs to be explained and supported in the paper, along with the list of which models have been tested to serve as AIRAs
- Detailed technique of evaluation of the AIRA is only presented in the supplementary material, but this is a core part of the paper and more empahsis should be given to it in the main pages of the paper. More information must be provided about human testing of the AIRA.


- some minor grammar and typing errors: see lines 041, 074, 079, 101, 118, 150,193,
- I believe the paper exceeds the page limit by a few pragraphs, so this needs to be addressed as well

**Questions:**

1. How should one choose AIRA models? Any guidance or recommendations or maybe a specific list to choose from?
2. What is -100% to +100% on the axis fo figure 6? could you add this explanation in the caption?
3. Any intuitive explanations why the Pearson correlation coeff. is so different for different values in Table 1?

---

### Official Review · Reviewer_djpi · 2024-11-04

**Soundness:** 3
**Presentation:** 3
**Contribution:** 2
**Rating:** 8
**Confidence:** 3

**Summary:**

The paper explores the concept of "first-person fairness" in chatbots, particularly focusing on biases that arise when chatbots respond based on user names, which may indicate gender or other demographic traits. The authors developed a tool using LLMs, called AI Research Assistant (AI RA), which aids in analyzing response patterns while protecting user privacy. The study found that while harmful biases were minimal (less than 1% of generated pairs), they were most notable in open-ended tasks, such as story generation. The research also highlighted that demographic biases might influence responses differently across groups.

**Strengths:**

- The topic of first-person fairness in chatbots is novel and could be important, especially for real-world applications.

- The paper provides an extensive evaluation of biases, using innovative privacy-preserving techniques.

- The "axis of differences" experiment was particularly compelling, showcasing the capability of language models to analyze data and identify patterns.

**Weaknesses:**

While I acknowledge the use of first-person fairness in contexts beyond large language models (LLMs), I find the distinction between this concept and third-person fairness in chatbots unclear. Since LLMs essentially sample from a distribution conditioned on the current input, incorporating the user's name into the system prompt effectively makes third-person and first-person fairness equivalent. In my view, intrinsically addressing third-person fairness should inherently resolve first-person fairness as well.

The work presents limited novelty. Although I recognize the value of the data used, the idea of evaluating gender bias in language models through the proxy of names is not new. The most interesting part of the study was the evaluation of patterns in LLM responses to different groups, and the authors introduced their LLM-based agent as the main contribution of this paper. However, as discussed in Section 3.1, this agent failed to successfully correlate with human ratings in this particular experiment.

Another concern is that the study used models from the same family to evaluate each other. This approach may not provide reliable bias evaluations, as models within the same family often share similar data, architectures, and underlying principles. Using models from different families would have offered more robust insights.

There are several presentation issues in the submission. For instance, there are multiple instances where spaces are missing between words or at sentence boundaries (e.g., Line 78: "Weuse"; Line 101: "Asin"; Line 118: "productmore"; Line 193: "Weextract"; Line 490: "Q2-smallas"; Lines 201 and 213: missing spaces between paragraph titles and the start of the paragraphs; and many other places where the first word of a sentence is not properly spaced from the period of the previous sentence, such as Line 110). Additionally, there is a faulty citation on Line 102. Moreover, in Section 2.1, from what I understand, y_1 and y_2 in the formula for h correspond to the variables response_1 and response_2 in the template shown in Figure 3. This is not immediately clear when the formula for h is explained in Lines 261–267, primarily due to insufficient detailed explanation of the variables in the formula.

**Questions:**

Q1: You mentioned that the user’s name is provided to the model in the context. Could you elaborate on how exactly this influences the model to differentiate between first-person and third-person fairness?

---

### Meta-Review · Area_Chair_Yc4e · 2024-12-19

**Metareview:**

**Summary:**

The authors conduct a study of chatbots' demographic biases towards users participating in conversations ("first-person fairness"). They introduce a LLM-based judge that can identify biased behavior, e.g. stereotyping, and provide explanations. They argue this can be an effective tool for scalability and privacy preservation in evaluation by eliminating the need for direct human observation. Results where the authors vary names mentioned in chatbot prompts indicate the proposed approach can effectively identify gender bias, while still struggling with other biases like racial bias.

**Strengths:**

- The authors study a large-scale chatbot system serving millions of chats with prompts from well-known chat datasets.

- The use of names as a proxy for demographic attribute leakage is realistic.

- The authors compare their LLM judge against diverse human judges' assessments of bias and show strong Pearson correlation.

- The explainability aspect of their framework can be used in making nuanced decisions about how to improve fairness of biased systems or identifying patterns in how bias manifests.

- There were concerns from the reviewers about the range of biases covered, however many well-known fairness papers focus on one dimension of bias (e.g. race or gender) and the coverage within this paper is actually relatively comprehensive. The authors also explicitly consider intersectional biases, which are understudied.

**Weaknesses:**

- The judge and the models being evaluated (beyond Table 1) are from the same family, potentially hindering generalizability.

- The technical contribution is fairly minimal, given that it relies mostly on prompt engineering and similar LLM-as-judge and open-ended generation bias evaluations have been done before.

- The chatbot data in the study is proprietary, limiting benefit to future research and reproducibility.

While the technical scope of this paper is limited, I think it could provide useful findings to the research community about scaling fairness evaluation. Therefore, I am leaning towards acceptance.

**Additional Comments On Reviewer Discussion:**

Most of the reviewers are strongly leaning towards acceptance, and the authors provided a detailed rebuttal and revision of potential over-claiming that the reviewers seem to agree has provided clarity on the paper's main motivation and definitions. They provide a reasonable explanation (proprietary data) for why they have not run equivalent experiments with other closed API models, but this does not explain the lack of open model results and this is a glaring omission in the paper. However, Table 1 with Llama and Anthropic comparison to human annotations of bias is a promising indicator that other models would perform similarly to the results in this paper. A limitation raised by the reviewers is the sole focus on names for demographic leakage, however I agree with the authors that this is a realistic setting to evaluate fairness.

---

### Decision · Program_Chairs · 2025-01-22

Accept (Spotlight)